# RangeAugment: Efficient Online Augmentation with Range Learning

## Abstract

State-of-the-art automatic augmentation methods (e.g., AutoAugment and RandAugment) for visual recognition tasks diversify training data using a large set of augmentation operations. The range of magnitudes of many augmentation operations (e.g., brightness and contrast) is continuous. Therefore, to make search computationally tractable, these methods use *fixed* and *manually-defined* magnitude ranges for each operation, which may lead to sub-optimal policies. To answer the open question on the importance of magnitude ranges for each augmentation operation, we introduce `RangeAugment` that allows us to efficiently learn the range of magnitudes for individual as well as composite augmentation operations. `RangeAugment` uses an auxiliary loss based on image similarity as a measure to control the range of magnitudes of augmentation operations. As a result, `RangeAugment` has a single scalar parameter for search, image similarity, which we simply optimize via linear search. `RangeAugment` integrates seamlessly with any model and learns model- and task-specific augmentation policies. With extensive experiments on the ImageNet dataset across different networks, we show that `RangeAugment` achieves competitive performance to state-of-the-art automatic augmentation methods with 4-5 times fewer augmentation operations. Experimental results on semantic segmentation and contrastive learning further shows `RangeAugment`'s effectiveness.

## 1 Introduction

Data augmentation is a widely used regularization method for training deep neural networks (LeCun et al., 1998; Krizhevsky et al., 2012; Szegedy et al., 2015; Perez & Wang, 2017; Steiner et al., 2021). These methods apply carefully designed augmentation (or image transformation) operations (e.g., color transforms) to increase the quantity and diversity of training data, which in turn helps improve the generalization ability of models. However, these methods rely heavily on expert knowledge and extensive trial-and-error experiments.

Recently, automatic augmentation methods have gained attention because of their ability to search for augmentation policy (e.g., combinations of different augmentation operations) that maximizes validation performance (Cubuk et al., 2019; 2020; Lim et al., 2019; Hataya et al., 2020; Zheng et al., 2021). In general, most augmentation operations (e.g., brightness and contrast) have two parameters: (1) the probability of applying them and (2) their range of magnitudes. These methods take a set of augmentation operations with a *fixed* (often discretized) range of magnitudes as an input, and produce a policy of applying some or all augmentation operations along with their parameters (Fig. 1). As an example, AutoAugment (Cubuk et al., 2019) discretizes the range of magnitudes and probabilities of 16 augmentation operations, and searches for sub-policies (i.e., composition of two augmentation operations along with their probability and magnitude) in a space of about $10^{32}$ possible combinations. These methods empirically show that automatic augmentation policies help improve performance of downstream networks. For example, AutoAugment improves the validation top-1 accuracy of ResNet-50 (He et al., 2016) by about 1.3% on the ImageNet dataset (Deng et al., 2009). In other words, these methods underline the importance of automatic composition of augmentation operations in improving validation performance. However, policies generated using these networks may be sub-optimal because they use hand-crafted magnitude ranges. The importance of magnitude ranges for each augmentation operation is still an open question. An obstacle in

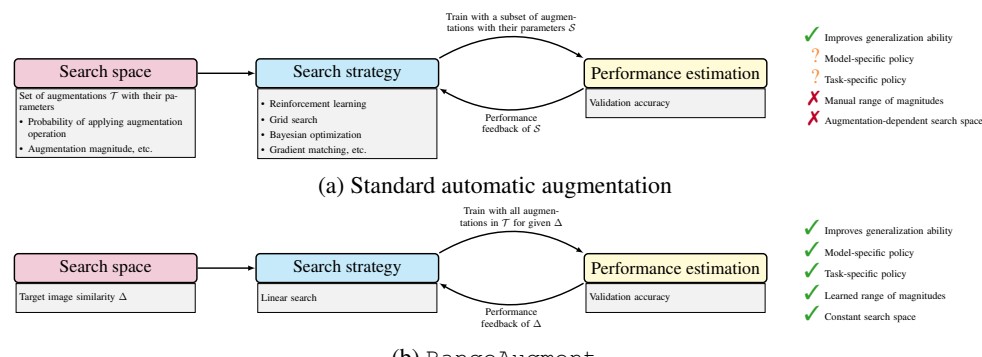

Figure 1: Comparison between `RangeAugment` and standard automatic augmentation methods. `RangeAugment`'s search space is independent of augmentation parameters, allowing us to learn model- and task-specific policies in a constant time.

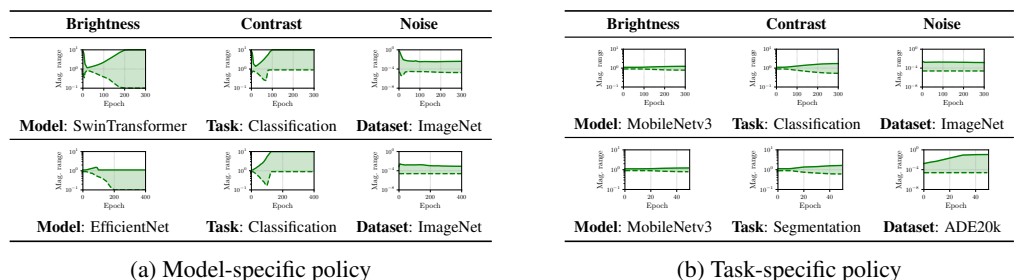

Figure 2: `RangeAugment` learns model- and task-specific policies. (a) shows the range of magnitudes for two different models on the same task and dataset while (b) shows the range of magnitudes for the same model on two different tasks. All models are trained end-to-end with `RangeAugment`. The target image similarity $\Delta$ (PSNR) is annealed from 40 to 5 in (a) and from 40 to 20 in (b).

answering this question is the range of magnitudes for most augmentation operations is continuous, which makes the search computationally intractable.

This paper introduces `RangeAugment`, a simple and efficient method to learn the range of magnitudes for each augmentation operation. Inspired by image similarity metrics (Hore & Ziou, 2010), `RangeAugment` introduces an auxiliary augmentation loss that allows us to learn the range of magnitudes for each augmentation operation. We realize this by controlling the similarity between the input and the augmented image for a given model and task. Rather than directly specifying the parameters for each augmentation operation, `RangeAugment` takes a target image similarity value as an input. The loss function is then formulated as a combination of the empirical loss and an augmentation loss. The objective of the augmentation loss is to match the target image similarity value. Therefore, the search objective in `RangeAugment` is to find the target similarity value that provides a good trade-off between minimizing the augmentation loss (i.e., matching the target similarity value) and the empirical loss. As a result, the augmentation policy learning in `RangeAugment` reduces to searching for a single scalar parameter, *target image similarity*, that maximizes downstream model's validation performance. We search for this target image similarity value via linear search. Empirically, we observe that this trade-off between the augmentation and empirical loss leads to better generalization ability of downstream model. Compared to existing automatic augmentation methods that require a large set of augmentation operations (usually 14-16 operations), `RangeAugment` is able to achieve competitive performance with only three simple operations (brightness, contrast, and additive Gaussian noise). Because `RangeAugment`'s search space is independent of augmentation parameters and is fully differentiable (Fig. 1), it can be trained end-to-end with any downstream model to learn model- and task-specific policies (Fig. 2).

We empirically demonstrate in Section 4 that `RangeAugment` allows us to learn model-specific policies when trained end-to-end with downstream models on the ImageNet dataset (Fig. 2a). Im-

portantly, `RangeAugment` achieves competitive performance to existing automatic augmentation methods (e.g., AutoAugment) with 4 to 5 times fewer augmentation operations. In Section 5, we apply `RangeAugment` to semantic segmentation and contrastive learning to demonstrate its simplicity and seamless integration ability to different tasks. We further show that `RangeAugment` learn task-specific policies (Fig. 2b). To the best of our knowledge, `RangeAugment` is the first automatic augmentation method that learns the range of magnitudes for each augmentation operation.

## 2 RELATED WORK

Data augmentation combines different augmentation operations (e.g., random brightness, random contrast, random Gaussian noise, and data mixing) to synthesize additional training data. Traditional augmentation methods rely heavily on expert knowledge and extensive trial-and-error experiments. In practice, these manual augmentation methods have been used to train different models on a variety of datasets and tasks (e.g., Szegedy et al., 2015; He et al., 2016; Zhao et al., 2017; Howard et al., 2019). However, these policies may not be optimal for all models.

Motivated by neural architecture search (Zoph & Le, 2017), recent methods focus on finding optimal augmentation policies automatically from data. AutoAugment formulates automatic augmentation as a reinforcement learning problem, and uses model's validation performance as a reward to find an augmentation policy leading to optimal validation performance. Because AutoAugment searches for several augmentation policy parameters, the search space is enormous and computationally intractable on large datasets and models. Therefore, in practice, policies found for smaller datasets are transferred to larger datasets. Since then, many follow-up works have focused on reducing the search space while delivering a similar performance to AutoAugment (Ratner et al., 2017; Lemley et al., 2017; LingChen et al., 2020; Li et al., 2020; Zheng et al., 2021; Liu et al., 2021a). The first line of research reduces the search time by introducing different hyper-parameter optimization methods, including population-based training (Ho et al., 2019), density matching (Lim et al., 2019; Hataya et al., 2020), and gradient matching (Zheng et al., 2021). The second line of research reduces the search space by making practical assumptions (Cubuk et al., 2020; Müller & Hutter, 2021). For instance, RandAugment (Cubuk et al., 2020) applies two transforms randomly with uniform probability. With these assumptions, RandAugment reduces AutoAugment's search space from $10^{32}$ to $10^2$ while maintaining downstream networks performance.

One common characteristic among these different automatic augmentation methods is that they use *fixed* and *manually-defined* range of magnitudes for different augmentation operations, and focus on diversifying training data by using a large set of augmentation operations (e.g., 14 transforms). This is because the range of magnitudes for most augmentation operations is continuous and large, and searching over this large range is practically infeasible. Unlike these works, `RangeAugment` focuses on *learning the magnitude range* of each augmentation operation (Figs. 1 and 2). We show in Section 4 that `RangeAugment` is able to learn model- and task-specific policies while delivering competitive performance to previous automatic augmentation methods across different models.

## 3 RANGEAUGMENT

Existing automatic augmentation methods search for composite augmentations over a large set of augmentation operations, but each augmentation operation has a manually-defined range of magnitudes. This paper introduces `RangeAugment`, a method for learning a range of magnitude for each augmentation operation (Fig. 3). `RangeAugment` uses image similarity between the input and augmented image to learn the range of magnitudes for each augmentation operation. In the rest of this section, we first formulate the problem (Section 3.1) and then elaborate on `RangeAugment`'s policy learning method (Section 3.2), followed by implementation details ( Section 3.3).

### 3.1 PROBLEM FORMULATION

Let $\mathcal{T} = \{T_1, \cdots, T_N\}$ be a set of $N$ differentiable augmentation operations. Each augmentation operation $T \in \mathcal{T}$ is parameterized by a scalar magnitude parameter $m \in \mathbb{R}$ such that $T(\cdot; m) : \mathcal{X} \to \mathcal{X}$ is defined on the image space $\mathcal{X}$. Let $\pi_\phi$ be an augmentation policy that defines a distribution over sub-policies $\mathcal{S} \sim \pi_\phi$ in `RangeAugment` such that $\mathcal{S} = \{T_i(\cdot; m_i)\}_{i=1}^N$. A

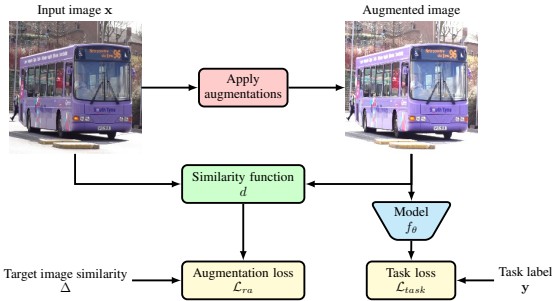

Figure 3: `RangeAugment`: End-to-end learning of augmentation policy with downstream model.

sub-policy $\mathcal{S}$ applies augmentation operations to an input image $\mathbf{x}$ with uniform probability as

$$\mathcal{S}(\mathbf{x}) := \mathbf{x}^{(N)}, \quad \mathbf{x}^{(i)} = T_i(\mathbf{x}^{(i-1)}; m_i), \qquad \mathbf{x}^{(0)} = \mathbf{x}. \tag{1}$$

For any given model and task, the goal of automatic augmentation is to find the augmentation policy $\pi_\phi$ that diversifies training data, and helps improve model's generalization ability the most. `RangeAugment` learns the range of magnitudes for each augmentation operation in $\mathcal{T}$. Formally, the policy parameters in `RangeAugment` are $\phi = \{(a_i, b_i)\}_{i=1}^N$ and the magnitude parameter $m_i \sim U(a_i, b_i)$ for the $i$-th augmentation operation in a sub-policy $\mathcal{S} = \{T_i(\cdot; m_i)\}_{i=1}^N$ is uniformly sampled, where $a_i \in \mathbb{R}$ and $b_i \in \mathbb{R}$ are learned parameters.

## 3.2 POLICY LEARNING

Diverse training data can be produced by using wider range of magnitudes, $(a_i, b_i)$, for each augmentation operation. However, directly searching for the optimal values of $(a_i, b_i)$ for each model and dataset is challenging because of its continuous nature. To address this, `RangeAugment` introduces an auxiliary loss which, in conjunction with the task-specific empirical loss, enables learning model-specific range of magnitudes for each augmentation operation in an end-to-end fashion.

Let $d : \mathcal{X} \times \mathcal{X} \to \mathbb{R}$ be a differentiable image similarity function that measures the similarity between the input and the augmented image. To control the range of magnitudes for each augmentation operation, `RangeAugment` minimizes the distance between the expected value of $d(\mathbf{x}, \mathcal{S}(\mathbf{x}))$ and a target image similarity value $\Delta \in \mathbb{R}$ using an augmentation loss function $\mathcal{L}_{\text{ra}}$ (e.g., smooth L1 loss or L2 loss). An example of $d$ and $\Delta$ are PSNR and target PSNR value respectively. When target PSNR value is small, the difference between the input and augmented image obtained after applying an augmentation operation (say brightness) will be large. In other words, for a smaller target PSNR value, the range of magnitudes for brightness operation will be wider and vice-versa.

For a given value of $\Delta$ and parameterized model $f_\theta$ with parameters $\theta$, the overall loss function to learn model- and task-specific augmentation policy is a weighted sum of the augmentation loss $\mathcal{L}_{\text{ra}}$ and the task-specific empirical loss $\mathcal{L}_{\text{task}}$:

$$\theta^*, \phi^* = \underset{\theta,\phi}{\arg\min}\, \mathbb{E}_{(\mathbf{x},\mathbf{y}) \sim \mathcal{D}_{\text{train}}} \left[ \mathbb{E}_{\mathcal{S} \sim \pi_\phi} \left[ \mathcal{L}_{\text{task}}(f_\theta(\mathcal{S}(\mathbf{x})), \mathbf{y}) + \lambda \mathcal{L}_{\text{ra}}(\mathbf{x}, \mathcal{S}(\mathbf{x}); \Delta) \right] \right], \tag{2}$$

where $\lambda$ and $\mathcal{D}_{\text{train}}$ represent weight term and training set respectively. Note that, in Eq. (2), re-parameterization trick on uniform distributions (Kingma & Welling, 2013) is applied to back-propagate through the expectation over $\mathcal{S} \sim \pi_\phi$.

The $\Delta$ in Eq. (2) allows `RangeAugment` to control diversity of augmented samples. Therefore, the augmentation policy learning in `RangeAugment` reduces to searching a single scalar parameter, $\Delta$, that maximizes downstream model's validation performance. `RangeAugment` finds the optimal value of $\Delta$ using a linear search.

## 3.3 IMPLEMENTATION DETAILS

We use PSNR as the image similarity function $d$ in our experiments because it is (1) a standard image quality metric, (2) differentiable, and (3) fast to compute. Across different downstream networks, we observe a 0.5%-3% training overhead over the empirical risk minimization baseline.

To find an optimal value of $\Delta$ in Eq. (2), we study two approaches: (1) fixed target PSNR ($\Delta \in \{5, 10, 20, 30\}$) and (2) target PSNR with a curriculum, where the value of $\Delta$ is annealed from 40 to $\delta$ and $\delta \in \{5, 10, 20, 30\}$. The learned ranges of magnitudes, $(a, b)$, can scale beyond the image space (e.g., negative values) and result in training instability. To prevent this, we clip the range of magnitudes if they are beyond extreme bounds of augmentation operations. We choose these extreme bounds such that objects in an image are hardly identifiable at or beyond the extreme points of the bounds (see Fig. 4).

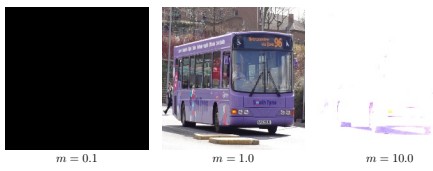

Figure 4: Example outputs of brightness operation, $T(\mathbf{x}; m) = m\mathbf{x}$, at different values of magnitude parameter $m$. At extremes (i.e., $m = 0.1$ or 10), the bus is hardly identifiable.

Also, because the focus of `RangeAugment` is to learn the range of magnitudes for each augmentation operation, we apply all augmentation operations in $\mathcal{T}$ with uniform probability.

To demonstrate the importance of magnitude ranges, we study `RangeAugment` with three basic operations (brightness, contrast, and additive Gaussian noise), and show empirically in Section 4 that `RangeAugment` can achieve competitive performance to existing methods with 4 to 5 times fewer augmentation operations.

## 4  EVALUATING RANGEAUGMENT ON IMAGE CLASSIFICATION

`RangeAugment` can learn model-specific augmentation policies. To evaluate this, we first study the importance of single and composite augmentation operations using ResNet-50 on the ImageNet dataset (Section 4.2). We then study model-level generalization of `RangeAugment` (Section 4.3).

### 4.1  EXPERIMENTAL SET-UP

**Dataset**  For image classification, we use the ImageNet dataset that has 1.28M training and 50k validation images spanning across 1000 categories. We use top-1 accuracy to measure performance.

**Baseline models**  To evaluate the effectiveness of `RangeAugment`, we study different CNN- and transformer-based models. We group these models into two categories based on their complexity: (1) *mobile:* MobileNetv1 (Howard et al., 2017), MobileNetv2 (Sandler et al., 2018), MobileNetv3 (Howard et al., 2019), and MobileViT (Mehta & Rastegari, 2021) and (2) *non-mobile:* ResNet-50 (He et al., 2016), ResNet-101, EfficientNet (Tan & Le, 2019), and SwinTransformer (Liu et al., 2021b). We implement `RangeAugment` using the CVNets library (Mehta et al., 2022) and use their baseline model implementations and training recipes. Additional experiments, including the effect of individual and joint learning on `RangeAugment`'s performance, along with training details are given in Appendices C and F.1 respectively.

**Baseline augmentation methods**  For an apples to apples comparison, each baseline model is trained with three different random seeds and the following baseline augmentation strategies: (1) ***Baseline*** - standard Inception-style pre-processing (random resized cropping and random horizontal flipping) (Szegedy et al., 2015), (2) ***RandAugment*** - Baseline pre-processing followed by RandAugment policy of Cubuk et al. (2020), (3) ***AutoAugment*** - Baseline pre-processing followed by AutoAugment policy of Cubuk et al. (2019), and (4) ***`RangeAugment`*** - Baseline pre-processing following by the proposed augmentation policy in Section 3.[1]

### 4.2  AUGMENTATION OPERATION CHARACTERIZATION USING RANGEAUGMENT

The quantity and diversity of training data can be increased by (1) increasing the magnitude range of each augmentation operation and (2) composite augmentation operations. Fig. 5 characterizes the

---

[1]As noted in Section 2, most previous works have focused on reducing the search time of AutoAugment while achieving similar performance (e.g., ResNet-50 on ImageNet: 77.6% (AutoAugment), 77.6% (RandAugment), and 77.6% (Fast AutoAugment)). Also, many state-of-the-art models (e.g., EfficientNet and SwinTransformer) have used either RandAugment or AutoAugment for data regularization. Therefore, to demonstrate the effectiveness of `RangeAugment`, we choose AutoAugment and RandAugment as baseline methods.

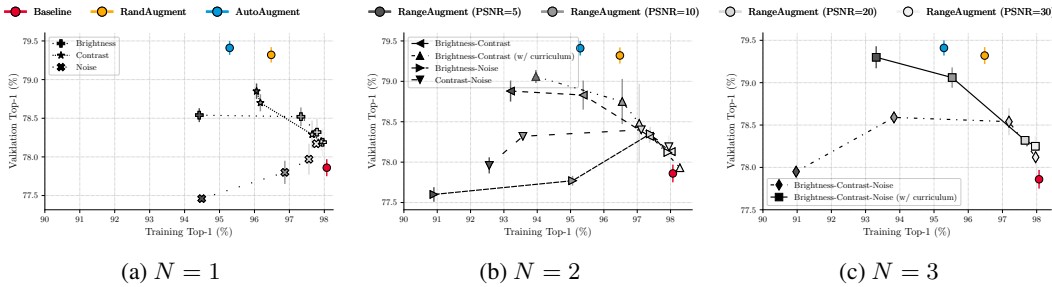

(a) $N = 1$      (b) $N = 2$      (c) $N = 3$

Figure 5: The performance of ResNet-50 on the ImageNet dataset when data diversity is increased by learning the range of magnitudes for single ($N = 1$) and composite augmentation operations ($N > 1$) using `RangeAugment`. For curriculum learning, target PSNR value is annealed from 40 to the value mentioned in the legend.

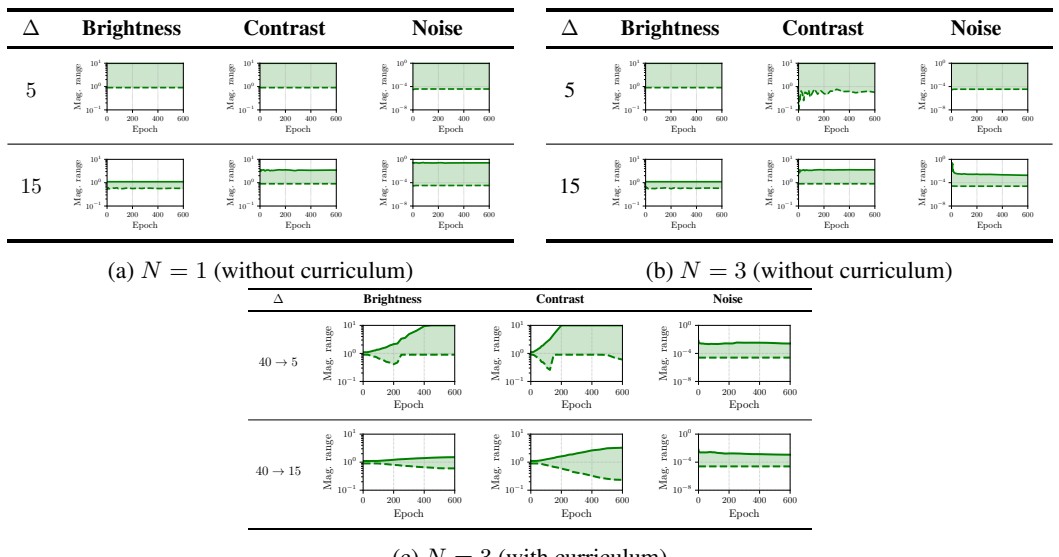

(a) $N = 1$ (without curriculum)      (b) $N = 3$ (without curriculum)

(c) $N = 3$ (with curriculum)

Figure 6: Learned range of magnitudes for different augmentation operations using `RangeAugment`. In (a) and (b), the target image similarity value (PSNR; $\Delta$) is fixed while in (c), $\Delta$ is annealed using cosine curriculum. The magnitude range in y-axis is in log scale.

effect of these variables on ResNet-50's performance on the ImageNet dataset. We can make the following observations:

1. Fig. 6a shows that single augmentation operation ($N = 1$) with wider magnitude ranges (e.g., the range of magnitudes at target PSNR of 5 are wider than the ones at target PSNR of 15) helped in improving ResNet-50's validation accuracy and reducing training accuracy, thereby improving its generalization capability (Fig. 5a). Particularly, increasing the magnitude range of contrast (or brightness) operation increased ResNet-50's validation performance over baseline by 1.0% (or 0.7%) while decreasing the training performance by 2% (or 3%). This is likely because wider magnitude ranges of an augmentation operation increases diversity of training data. On the other hand, the additive Gaussian noise operation with a wider magnitude range slightly dropped the validation accuracy, but still reduces the training accuracy. In other words, it improves ResNet-50's generalization ability.

2. Composite augmentation operations ($N > 1$; Figs. 5b and 5c) reduces the training accuracy significantly while having a validation performance similar to single augmentation operation. This is expected as composite operations further increases training data diversity.

3. Fig. 6c shows that progressively learning to increase the diversity of augmented samples (i.e., narrower to wider magnitude ranges[2]) using a cosine curriculum further improves the performance (Fig. 5c). A plausible explanation is that the learned magnitude ranges of different augmentation operations using RangeAugment with fixed target PSNR may get stuck near poor solutions. Because the range of magnitudes are wider for these solutions (e.g., the range of magnitudes at target PSNR value of 5 in Fig. 6a & Fig. 6b), RangeAugment samples more diverse data from the beginning of the training, making training difficult. Gradually annealing target PSNR from high to low (e.g., 40 to 5 in Fig. 6c) allows RangeAugment to increase the data diversity slowly, thereby helping model to learn better representations. Importantly, it also allows RangeAugment to identify useful ranges for each augmentation operation. For example, the range of magnitudes for the additive Gaussian noise operation is relatively narrower when optimizing with curriculum (e.g., annealing target PSNR from 40 to 5; Fig. 6c) compared to optimizing with a fixed target PSNR of 5 (Fig. 6b). This indicates that noise operation with narrow magnitude range is favorable for training ResNet-50 on the ImageNet classification task. This concurs with results in Figs. 5a and 5b where we observed that noise operation does not improve ResNet-50's validation performance. Moreover, our findings with progressively increasing data diversity are consistent with previous works (e.g., Bengio et al., 2009; Tan & Le, 2021) that shows scheduling training samples from easy to hard helps improve model's performance.

   Interestingly, ResNet-50 with RangeAugment ($N = 3$) achieves comparable validation accuracy and a smaller generalization gap as compared to state-of-the-art methods which use more augmentation operations ($N > 14$) to increase data diversity during training. We conjecture that the difference in the generalization gap between existing methods and RangeAugment is probably caused by insufficient policy search in existing methods as they use manually-defined magnitude ranges for each augmentation operation during search.

> **Observation 1:** Composite augmentation operations with wider magnitude ranges is important for improving downstream model's generalization ability.

In the rest of experiments, we will use all three augmentations ($N = 3$) with cosine curriculum.

## 4.3 MODEL-LEVEL GENERALIZATION OF RANGEAUGMENT

Fig. 5 shows RangeAugment is effective for ResNet-50. Natural questions that arise are:

1. *Can RangeAugment be applied to other vision models?* RangeAugment's seamless integration ability with little training overhead (0.5% to 3%) over the baseline model allows us to study the generalization capability of different vision models easily. Fig. 7 shows the performance of different models with RangeAugment. When data regularization is increased for mobile models by decreasing the target PSNR value from 40 to 5, the training as well as validation accuracy of different mobile models is decreased significantly as compared to the baseline. This is likely because of the limited capacity of these models. On the other hand, data regularization improved the performance of non-mobile models significantly. Consistent with our observations for ResNet-50 in Fig. 5, we found that non-mobile models trained with RangeAugment are able to achieve competitive performance to state-of-the-art automatic augmentation methods, such as AutoAugment ($N = 16$), but with fewer augmentations ($N = 3$).

2. *Does RangeAugment learn architecture-specific augmentations?* Fig. 2a shows the learned magnitude ranges for different augmentation operations for a transformer- (SwinTransformer) and a CNN-based (EfficientNet) model. Though both of these architectures use the same curriculum in RangeAugment (i.e., target PSNR is annealed from 40 to 5), they learn different magnitude ranges for each augmentation operation. This shows that RangeAugment is capable of learning model-specific magnitude ranges for each augmentation operation.

3. *Does RangeAugment increase variance on model performance?* Because of the stochastic training and presence of randomness during different stages of training including RangeAugment, there may be some variability in model's performance. To measure the variability in model's performance, we run each experiment with three different random seeds. For different models, the standard deviation of model's validation accuracy is between 0.01 and 0.2,

---

[2]Wider magnitude ranges produce diverse augmented samples and vice versa (Appendix D).

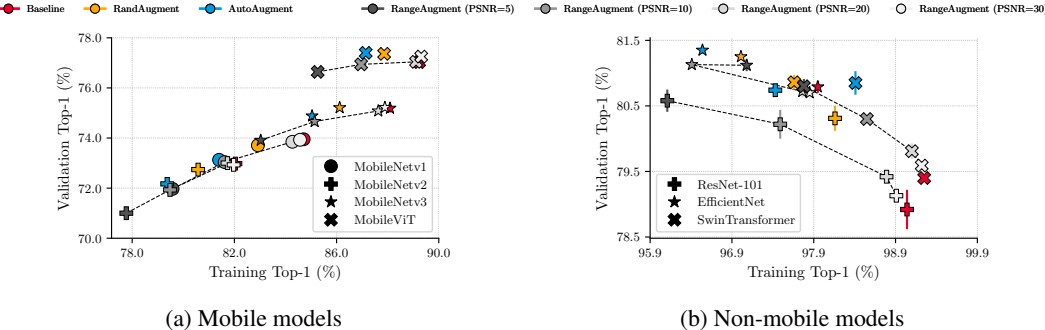

(a) Mobile models           (b) Non-mobile models

Figure 7: Performance of different models on the ImageNet dataset using `RangeAugment`. Here, the target PSNR value is annealed from 40 to the value mentioned in the legend.

and is in accordance with previous works on the ImageNet dataset (Radosavovic et al., 2020; Wightman et al., 2021). These results show that training models with `RangeAugment` leads to a stable model performance.

---

**Observation 2:** Non-mobile models benefit from data regularization. On the ImageNet dataset, we recommend to train non-mobile models using a curriculum that anneals $\Delta$ from high (e.g., target PSNR=40) to low (e.g., target PSNR=5 or 10) similarity between input and augmented images.

---

## 5 TASK-LEVEL GENERALIZATION OF RANGEAUGMENT

Section 4 shows the effectiveness of `RangeAugment` on different downstream models on the ImageNet dataset. However, one might ask whether `RangeAugment` can be used for tasks other than image classification. To evaluate this, we study `RangeAugment` with two tasks, semantic segmentation (Section 5.1) and contrastive image-language pre-training (Section 5.2).

### 5.1 SEMANTIC SEGMENTATION ON THE ADE20K DATASET

**Dataset and baseline models** We use ADE20k dataset (Zhou et al., 2017) that has 20k training and 2k validation images across 150 semantic classes. We report the segmentation performance in terms of mean intersection over union (mIoU) on the validation set.

We integrate mobile and non-mobile classification models with the Deeplabv3 segmentation head (Chen et al., 2018a) and finetune each model for 50 epochs. See Appendix F.1 for training details. We do not study SwinTransformer for semantic segmentation because it is not compatible with Deeplabv3's segmentation head design as it adjusts the atrous rate of convolutions to control the output stride of backbone network.

**Baseline augmentation methods** For semantic segmentation, experts have hand-crafted augmentation policies, and these manual policies are used to train state-of-the-art semantic segmentation methods (e.g., Chen et al., 2017; Zhao et al., 2017; Xie et al., 2021; Liu et al., 2021b). For an apples to apples comparison, we train each segmentation model with three different random seeds and compare with the following baselines: (1) ***Baseline*** - standard pre-processing (randomly resize short image dimension, random horizontal flip, and random crop), (2) ***Manual*** - baseline pre-processing with hand-crafted augmentation operations (color jittering using photometric distortion, random rotation, and random Gaussian noise), and (3) ***RangeAugment*** - baseline pre-processing with learnable range of magnitudes for brightness, contrast, and noise (i.e., $N = 3$). For reference, we include DeepLabv3 results (if available) from a popular segmentation library (MMSegmentation, 2020).

**Results** Table 1 shows that `RangeAugment` improves the performance of different models in comparison to other augmentation methods. Interestingly, for semantic segmentation, the magnitude range of additive Gaussian noise is wider compared to image classification (Fig. 2b). This concurs with previous manual augmentation methods which also found that Gaussian noise is important for

| Backbone | Augmentation method | | | | | | MMSegmentation |
| | Baseline | Manual | RangeAugment (Ours) | | | | (reference) |
| | | | (40, 5) | (40, 10) | (40, 20) | (40, 30) | |
|---|---|---|---|---|---|---|---|
| MobileNetv1 | $38.77 \pm 0.20$ | $38.12 \pm 0.16$ | $36.46 \pm 0.19$ | $38.20 \pm 0.26$ | $38.96 \pm 0.34$ | $\mathbf{39.37 \pm 0.19}$ | – |
| MobileNetv2 | $37.74 \pm 0.29$ | $37.10 \pm 0.27$ | $35.58 \pm 0.30$ | $37.43 \pm 0.73$ | $38.06 \pm 0.17$ | $\mathbf{38.23 \pm 0.36}$ | 34.08 |
| MobileNetv3 | $37.58 \pm 0.67$ | $36.68 \pm 0.34$ | $34.80 \pm 0.16$ | $36.54 \pm 0.12$ | $37.77 \pm 0.24$ | $\mathbf{38.10 \pm 0.13}$ | – |
| MobileViT | $37.69 \pm 0.50$ | $37.19 \pm 0.47$ | $35.04 \pm 0.83$ | $36.70 \pm 0.19$ | $37.94 \pm 0.45$ | $\mathbf{38.41 \pm 0.22}$ | – |
| ResNet-50 | $42.27 \pm 0.54$ | $43.29 \pm 0.27$ | $41.56 \pm 0.21$ | $42.95 \pm 0.22$ | $\mathbf{43.31 \pm 0.16}$ | $43.00 \pm 0.28$ | 42.42 |
| ResNet-101 | $43.29 \pm 0.17$ | $44.04 \pm 0.52$ | $43.22 \pm 0.52$ | $43.89 \pm 0.43$ | $\mathbf{44.77 \pm 0.28}$ | $43.95 \pm 0.31$ | 44.08 |
| EfficientNet | $40.86 \pm 0.55$ | $41.15 \pm 0.65$ | $39.42 \pm 0.29$ | $40.39 \pm 0.48$ | $\mathbf{41.43 \pm 0.36}$ | $41.08 \pm 0.36$ | – |

Table 1: Semantic segmentation on the ADE20k dataset.

| Model | Dataset | Test resolution | Top-1 Accuracy (%) |
|---|---|---|---|
| CLIP of Radford et al. (2021) | Proprietary | $224 \times 224$ | 68.3%[†] |
| OpenCLIP of Ilharco et al. (2021) | LAION-400M | $224 \times 224$ | 67.1% |
| CLIP w/ RangeAugment (Ours) | LAION-400M | $160 \times 160$ | 65.8% |
| | | $192 \times 192$ | 67.4% |
| | | $224 \times 224$ | 68.3% |
| | | $256 \times 256$ | 68.8% |
| | | $320 \times 320$ | 69.1% |

Table 2: Zero-shot performance on ImageNet. Each entry of CLIP with RangeAugment is the same model, but evaluated at different resolutions. [†] Results of CLIP with the same language prompts as OpenCLIP.

semantic segmentation (Zhao et al., 2017; Asiedu et al., 2022). Overall, these results suggest that RangeAugment learns task-specific augmentation policies.

## 5.2 CONTRASTIVE IMAGE-LANGUAGE PRE-TRAINING ON THE LAION-400M DATASET

**Dataset and baseline models** We crawl the LAION-400M dataset (Schuhmann et al., 2021) and download about 304M image-language pairs, which are then used for pre-training. We report zero-shot top-1 accuracy on ImageNet's validation set and use the same language prompts as OpenCLIP (Ilharco et al., 2021).

We train CLIP (Radford et al., 2021) with RangeAugment from scratch (Appendix F.1). The model uses ViT-B/16 (Dosovitskiy et al., 2020) as its image encoder and transformer as its text encoder, and minimizes contrastive loss during training. We use multi-scale sampler of Mehta et al. (2022) to make CLIP more robust to input scale changes. Because less data regularization is required at a scale of 100M+ samples (Radford et al., 2021; Zhai et al., 2022), we anneal the target PSNR in RangeAugment from 40 to 20. We compare the performance with CLIP and OpenCLIP.

**Results** Table 2 compares the zero-shot performance of different models. For the same training dataset and zero-shot language prompts, RangeAugment delivers 1.2% better performance than OpenCLIP at an inference resolution of $224 \times 224$.

**Observation 3:** RangeAugment learns model- and task-specific augmentation policies.

## 6 CONCLUSION

This paper introduces an end-to-end method for learning model- and task-specific automatic augmentation policies with a constant search time. We demonstrated that RangeAugment delivers competitive performance to existing methods across different downstream models on the image classification task. This is despite the fact that RangeAugment uses only three basic augmentation operations as opposed to a large set of complex augmentation operations in existing methods. These results underline the importance of magnitude range of augmentation operations in automatic augmentation. We also showed that RangeAugment can be seamlessly integrated with other tasks and achieve similar or better performance than existing methods. In the future, we plan to apply RangeAugment to learn the range of magnitudes for complex augmentation operations (e.g., geometric transformations) using different image similarity functions (e.g., SSIM). In addition to learning the range of magnitudes of each augmentation operation, we plan to apply RangeAugment to learn how to compose different augmentation operations with a constant search time.

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

## A  COMPARISON WITH EXISTING METHODS

**ImageNet classification**  State-of-the-art methods incorporate random erasing (Zhong et al., 2020), mixup transforms (Zhang et al., 2017; Yun et al., 2019) in addition to automatic augmentation methods (e.g., RandAugment and AutoAugment). Table 3 shows that models trained with `RangeAugment` are able to achieve similar or better performance than existing automatic augmentation methods with $4 - 5\times$ more augmentation operations.

| Model | Source | Data augmentation methods | | | Top-1 |
|---|---|---|---|---|---|
| | | Auto aug. method | Random erase | Mixup | accuracy |
| **Mobile models** | | | | | |
| **MobileNetV1-1.0** | Orig. (Howard et al., 2017) | ✗ | ✗ | ✗ | 70.6% |
| | Our repro. | RandAugment ($N = 14$) | ✗ | ✗ | **73.7%** |
| | Our repro. | AutoAugment ($N = 16$) | ✗ | ✗ | 73.1% |
| | Ours | RangeAugment ($N = 3$) | ✗ | ✗ | **73.8%** |
| **MobileNetV2-1.0** | Orig. (Sandler et al., 2018) | ✗ | ✗ | ✗ | 72.0% |
| | Our repro. | RandAugment ($N = 14$) | ✗ | ✗ | 72.7% |
| | Our repro. | AutoAugment ($N = 16$) | ✗ | ✗ | 72.1% |
| | Ours | RangeAugment ($N = 3$) | ✗ | ✗ | **73.0%** |
| **MobileNetV3-Large** | Orig. (Howard et al., 2019) | ✗ | ✗ | ✗ | 74.6% |
| | Our repro. | RandAugment ($N = 14$) | ✗ | ✗ | **75.2%** |
| | Our repro. | AutoAugment ($N = 16$) | ✗ | ✗ | 74.9% |
| | Ours | RangeAugment ($N = 3$) | ✗ | ✗ | **75.1%** |
| **MobileViT-Small** | Orig. (Mehta & Rastegari, 2021) | ✗ | ✗ | ✗ | **78.4%** |
| | Our repro. | RandAugment ($N = 14$) | ✗ | ✗ | 77.4% |
| | Our repro. | AutoAugment ($N = 16$) | ✗ | ✗ | 77.4% |
| | Ours | RangeAugment ($N = 3$) | ✗ | ✗ | **78.2%** |
| **Non-mobile models** | | | | | |
| **ResNet-50** | Orig. (He et al., 2016) | ✗ | ✗ | ✗ | 76.2% |
| | TIMM (Wightman et al., 2021) | RandAugment ($N = 14$) | ✗ | ✓ | **80.4%** |
| | Ours | RangeAugment ($N = 3$) | ✗ | ✓ | **80.2%** |
| **ResNet-101** | Orig. (He et al., 2016) | ✗ | ✗ | ✗ | 77.4% |
| | TIMM (Wightman et al., 2021) | RandAugment ($N = 14$) | ✗ | ✓ | 81.5% |
| | Ours | RangeAugment ($N = 3$) | ✗ | ✓ | **82.0%** |
| **EfficientNet-B0** | Orig. (Tan & Le, 2019) | AutoAugment ($N = 16$) | ✓ | ✓ | **77.1%** |
| | TIMM (Wightman et al., 2021) | RandAugment ($N = 14$) | ✗ | ✓ | 77.0% |
| | Ours | RangeAugment ($N = 3$) | ✗ | ✓ | **77.3%** |
| **EfficientNet-B1** | Orig. (Tan & Le, 2019) | AutoAugment ($N = 16$) | ✓ | ✓ | 79.1% |
| | TIMM (Wightman et al., 2021) | RandAugment ($N = 14$) | ✗ | ✓ | 79.2% |
| | Ours | RangeAugment ($N = 3$) | ✗ | ✓ | **79.5%** |
| **EfficientNet-B2** | Orig. (Tan & Le, 2019) | AutoAugment ($N = 16$) | ✓ | ✓ | 80.1% |
| | TIMM (Wightman et al., 2021) | RandAugment ($N = 14$) | ✗ | ✓ | 80.4% |
| | Ours | RangeAugment ($N = 3$) | ✗ | ✓ | **81.3%** |
| **EfficientNet-B3** | Orig. (Tan & Le, 2019) | AutoAugment ($N = 16$) | ✓ | ✓ | 81.6% |
| | TIMM (Wightman et al., 2021) | RandAugment ($N = 14$) | ✗ | ✓ | 81.4% |
| | Ours | RangeAugment ($N = 3$) | ✗ | ✓ | **81.9%** |
| **Swin-Tiny** | Orig. (Liu et al., 2021b) | RandAugment ($N = 14$) | ✓ | ✓ | **81.3%** |
| | Ours | RangeAugment ($N = 3$) | ✗ | ✓ | **81.1%** |
| **Swin-Small** | Orig. (Liu et al., 2021b) | RandAugment ($N = 14$) | ✓ | ✓ | **83.0%** |
| | Ours | RangeAugment ($N = 3$) | ✗ | ✓ | **82.8%** |

Table 3: **Accuracy comparison of different models trained with different methods on the ImageNet validation set.** `RangeAugment` with simple and $4 - 5\times$ fewer transforms is able to deliver similar or better performance to state-of-the-art methods with complex automatic augmentation policies. For mobile models, we decay $\Delta$ from 40 to 30 while for non-mobile models, we decay $\Delta$ from 40 to 5 (as per observations in Section 4.1). Methods whose performance is within the standard deviation range of $\pm 0.2$ of the best model are highlighted in bold. Note that RandAugment in TIMM is a custom implementation that delivers better performance than the RandAugment of Cubuk et al. (2020), and is widely used for training recent classification networks on the ImageNet, including SwinTransformers. Here, $N$ denotes the number of augmentation operations.

**Semantic segmentation on ADE20k**  Table 4 compares the performance of different segmentation architectures for the same backbone. Compared to highly-tuned augmentation recipes of MMSeg (MMSegmentation, 2020) and CSAIL (Zhou et al., 2017) segmentation libraries, `RangeAugment` is able to achieve better performance consistently across different backbones.

Furthermore, Table 5 shows that DeepLabv3 with ResNet-101 backbone, when trained with `RangeAugment`, delivers the same performance as UPerNet (Xiao et al., 2018) with SwinTransformer (Liu et al., 2021b) and ConvNext (Liu et al., 2022) backbones while being $3\times$ FLOP efficient.

| Backbone | Seg. architecture | Source | mIOU |
|---|---|---|---|
| MobileNetv2 | DeepLabv3 (Chen et al., 2017) | MMSeg | 34.0 |
| | PSPNet (Zhao et al., 2017) | CSAIL | 35.8 |
| | DeepLabv3 (Chen et al., 2017) | Our repro. | 37.9 |
| | DeepLabv3 (Chen et al., 2017) w/ RangeAugment (Ours) | | **38.6** |
| ResNet-50 | UPerNet | CSAIL | 40.4 |
| | UPerNet (Xiao et al., 2018) | MMSeg | 42.1 |
| | PSPNet (Zhao et al., 2017) | MMSeg | 42.5 |
| | DeepLabv3 (Chen et al., 2017) | MMSeg | 42.7 |
| | DeepLabv3 (Chen et al., 2017) | Our repro. | 43.0 |
| | DeepLabv3 (Chen et al., 2017) w/ RangeAugment (Ours) | | **44.0** |
| ResNet-101 | UPerNet (Xiao et al., 2018) | CSAIL | 42.0 |
| | PSPNet (Zhao et al., 2017) | MMSeg | 42.5 |
| | UPerNet (Xiao et al., 2018) | MMSeg | 43.8 |
| | DeepLabv3 (Chen et al., 2017) | MMSeg | 45.0 |
| | DeepLabv3 (Chen et al., 2017) | Our repro. | 45.2 |
| | DeepLabv3 (Chen et al., 2017) w/ RangeAugment (Ours) | | **46.5** |

Table 4: Comparison between different state-of-the-art segmentation method for the same backbone. Models trained with RangeAugment is able to deliver better performance than highly-tuned augmentation pipelines in popular segmentation libraries (CSAIL (Zhou et al., 2017) and MMSeg (MMSegmentation, 2020)).

| Seg. model | # Params. | FLOPs | mIOU |
|---|---|---|---|
| Swin w/ UPerNet (Liu et al., 2021b) | **60 M** | 945 G | 45.8 |
| ConvNext w/ UPerNet (Liu et al., 2022) | **60 M** | 939 M | **46.7** |
| ResNet-101 w/ DeepLabv3 & RangeAugment (Ours) | 77 M | **303 G** | 46.5 |

Table 5: **DeepLabv3 with RangeAugment delivers similar performance to UPerNet while being $3\times$ more FLOP efficient**. RangeAugment improved the segmentation accuracy of ResNet-101 with DeepLabv3 significantly; delivering competitive performance to state-of-the-art segmentation model, UPerNet, with recent backbones (SwinTransformer- and ConvNext).

Overall, these segmentation results underline the effectiveness of RangeAugment.

## B  TRANSFERRING AUGMENTATION POLICY

Searching model- and task-specific policy may be expensive. Therefore, a common practice is to transfer the policy found on one dataset to another. This section evaluates if the augmentation curriculum of RangeAugment can be used across different tasks and datasets. We compare the accuracy of RangeAugment with publicly reproduced models as there performance is often better than those reported in the paper. For experiments in this section, we follow our observations in Section 5 and anneal the PSNR value $\Delta$ from 40 to 20.

**Object detection on COCO**  Following previous works, we use ResNet-50 as a backbone and train Mask R-CNN on the COCO dataset (Lin et al., 2014). Table 6 shows that RangeAugment improves the detection accuracy of Mask-RCNN significantly.

| Model | Optim. updates | BBOX mAP |
|---|---|---|
| Detectron2 | 270k | 41.0 |
| MMDetection | 270k | 40.9 |
| RangeAugment (Ours) | **70k** | 41.0 |
| RangeAugment (Ours) | 230k | **42.6** |

Table 6: **Enhanced object detection results** of Mask R-CNN with RangeAugment on COCO. Following a standard convention for reporting object detection performance of Mask R-CNN, we also report the number of optimization updates (or schedule). We use similar hyper-parameters, including learning rate, as Detectron2 (Wu et al., 2019) and MMDetection (Chen et al., 2019).

**Semantic segmentation on PASCAL VOC 2012**  Following previous segmentation methods, we use ResNet-101 as a backbone and train DeepLabv3 on the PASCAL VOC 2012 dataset (Everingham et al., 2012). Table 7 shows that DeepLabv3 with RangeAugment attains the best performance.

| Seg. model | mIoU |
|---|---|
| ANN (Zhu et al., 2019) | 76.7 |
| APCNet (He et al., 2019) | 78.5 |
| CCNet (Huang et al., 2019) | 77.9 |
| DeepLabv3 (Chen et al., 2017) | 77.9 |
| DeepLabv3+ (Chen et al., 2018b) | 78.6 |
| PSPNet (Zhao et al., 2017) | 78.5 |
| UPerN (Xiao et al., 2018) | 77.4 |
| DeepLabv3 w/ RangeAugment (Ours) | **84.0** |

Table 7: **Comparison with state-of-the-art semantic segmentation methods** with ResNet-101 backbone on the PASCAL VOC validation set. We do not use multi-scale evaluation. The results of different segmentation models are from MMSegmentation (2020). Also, our training recipes, including batch size and learning rate, are similar to MMSegmentation (2020).

## C    ABLATIONS ON THE IMAGENET DATASET

In this section, we study different components of RangeAugment using ResNet-50. For learning augmentation policy, we anneal the target image similarity (PSNR) value $\Delta$ from 40 to 5.

**Effect of different curriculum**    We trained RangeAugment with two curriculum's: (1) linear and (2) cosine. We found that cosine curriculum delivers 0.1-0.2% better performance than linear. Therefore, we use cosine curriculum.

**Effect of $\lambda$**    The weight term, $\lambda$, in Eq. (2) allows RangeAugment to balance the trade-off between augmentation loss $\mathcal{L}_{\text{ra}}$ and empirical loss $\mathcal{L}_{\text{task}}$. To study its impact, we vary the value of $\lambda$ from 0.0 to 0.15. Empirical results in Fig. 8 shows that the good range for $\lambda$ is between 0.0006 and 0.002. In our experiments, we use $\lambda = 0.0015$.

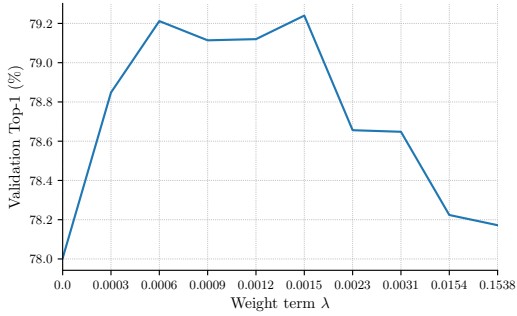

Figure 8: Effect of weight term, $\lambda$, on ResNet-50's performance on the ImageNet dataset.

**Effect of joint vs. independent optimization**    An expected behavior for learning model-specific augmentation policy using RangeAugment is that task-specific loss $\mathcal{L}_{\text{task}}$ in Eq. 2 should contribute towards policy learning. To validate it, ResNet-50 is trained independently[3] as well as jointly on the ImageNet dataset. We found that the top-1 accuracy of ResNet-50 dropped by about 1% when it is trained independently. This is likely because independent training allowed RangeAugment to produce augmented images with more additive Gaussian noise (Fig. 9), resulting in performance drop. This concurs with our observations in Section 4, especially Figs. 5 and 6, where we found that sampling augmented images from wider magnitude range for additive Gaussian noise operation dropped ResNet-50's performance on the ImageNet dataset. A plausible explanation is that PSNR is more sensitive to noise operation (Hore & Ziou, 2010), allowing RangeAugment to learn wider magnitude ranges for noise operation when trained independently as compared to joint training. Overall, these results suggest that joint training helps in learning model-specific policy.

---

[3]The augmented image is detached before feeding to the model.

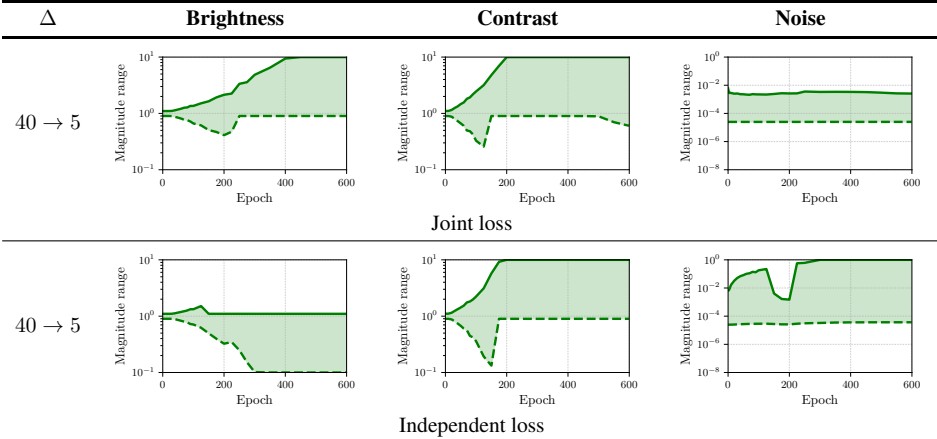

Figure 9: The effect of learning magnitude ranges by jointly optimizing the loss terms $\mathcal{L}_{ra}$ and $\mathcal{L}_{task}$ (top row) compared to only optimizing $\mathcal{L}_{ra}$ (bottom row). Training ResNet-50 with the joint loss leads to smaller magnitudes of noise, and improves validation accuracy by approximately $1\%$ on the ImageNet dataset.

## D  VISUALIZATION OF AUGMENTED SAMPLES

Fig. 10 visualizes augmented samples produced by `RangeAugment` at different stages of training ResNet-50 with curriculum. We can see that the range of magnitudes (Fig. 10a) and diversity of augmented samples (Fig. 10b) increases as training progresses.

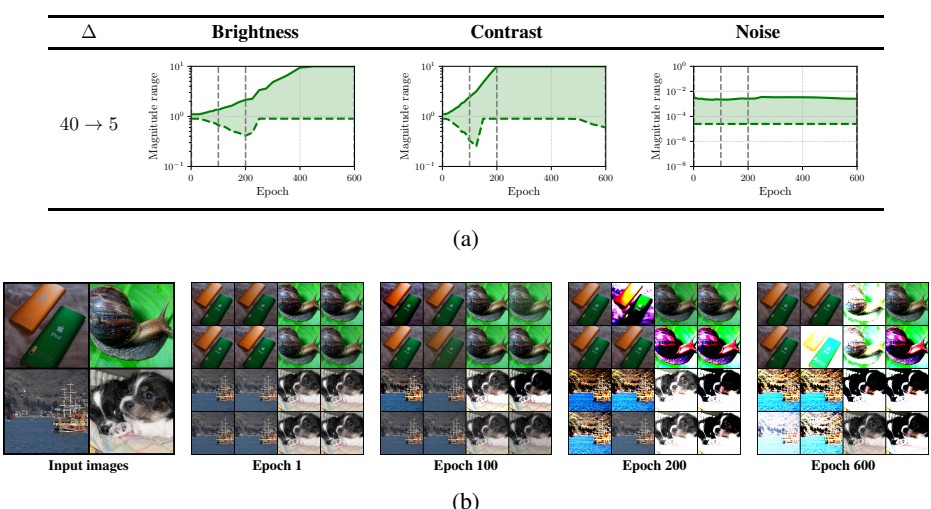

Figure 10: Visualization of augmented samples. (a) Learned magnitude ranges of different augmentations when ResNet-50 is trained jointly with `RangeAugment` using a cosine curriculum. (b) Four image samples visualized at different epoch intervals. For illustration purposes, we visualize four random augmented samples produced by `RangeAugment` for each image.

## E  LEARNED MAGNITUDE RANGES FOR CLIP WITH RANGEAUGMENT

Fig. 11 shows the learned magnitude ranges of different augmentation operations for training the CLIP model with `RangeAugment`. Unlike image classification (Section 4) and semantic segmentation (Section 5.1) results, the CLIP model uses little augmentation. This is expected because the LAION dataset is orders of magnitude larger than classification and segmentation datasets.

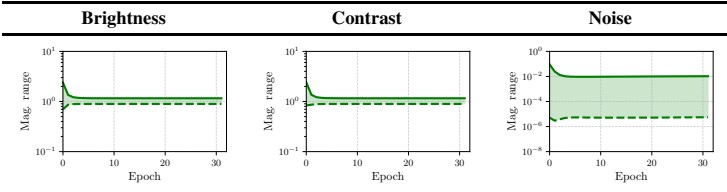

Figure 11: Learned magnitude ranges of different augmentations when the CLIP model is trained with `RangeAugment` using a cosine curriculum. Target PSNR $\Delta$ is annealed from 40 to 20.

## F TRAINING DETAILS

### F.1 TRAINING HYPER-PARAMETERS

Table 8 summarizes training recipe used for training different models across different tasks.

| | Non-mobile models | | | | Mobile models | | | |
|---|---|---|---|---|---|---|---|---|
| | **ResNet-50** | **ResNet-101** | **EfficientNet** | **SwinTransformer** | **MobileViT** | **MobileNetv1** | **MobileNetv2** | **MobileNetv3** |
| Epochs | 600 | 600 | 400 | 300 | 300 | 300 | 300 | 300 |
| Batch size | 1024 | 1024 | 2048 | 1024 | 1024 | 512 | 1024 | 2048 |
| Data sampler | MSc-VBS | MSc-VBS | MSc-VBS | SSc-FBS | MSc-VBS | MSc-VBS | MSc-VBS | MSc-VBS |
| Max. LR | 0.4 | 0.4 | 0.8 | $10^{-3}$ | $2 \times 10^{-3}$ | 0.4 | 0.4 | 0.8 |
| Min. LR | $2 \times 10^{-4}$ | $2 \times 10^{-4}$ | $4 \times 10^{-4}$ | $10^{-5}$ | $2 \times 10^{-4}$ | $2 \times 10^{-4}$ | $2 \times 10^{-4}$ | $4 \times 10^{-4}$ |
| Warmup init. LR | 0.05 | 0.05 | 0.1 | $10^{-6}$ | $2 \times 10^{-4}$ | 0.05 | 0.05 | 0.1 |
| Warmup epochs | 5 | 5 | 5 | 20 | 16 | 3 | 6 | 5 |
| LR Annealing | cosine | cosine | cosine | cosine | cosine | cosine | cosine | cosine |
| Weight decay | $4 \times 10^{-5}$ | $4 \times 10^{-5}$ | $4 \times 10^{-5}$ | $5 \times 10^{-2}$ | 0.01 | $4 \times 10^{-5}$ | $4 \times 10^{-5}$ | $4 \times 10^{-5}$ |
| Optimizer | SGD | SGD | SGD | AdamW | AdamW | SGD | SGD | SGD |
| Momentum | 0.9 | 0.9 | 0.9 | ✗ | ✗ | 0.9 | 0.9 | 0.9 |
| Label smoothing $\epsilon$ | 0.1 | 0.1 | 0.1 | 0.1 | 0.1 | 0.1 | 0.1 | 0.1 |
| Stoch. Depth | ✗ | ✗ | ✗ | 0.3 | ✗ | ✗ | ✗ | ✗ |
| Grad. clipping | ✗ | ✗ | ✗ | 5.0 | ✗ | ✗ | ✗ | ✗ |
| # parameters | 25.6 M | 44.5 M | 12.3 M | 49.6 M | 5.6 M | 4.2 M | 3.5 M | 5.4 M |
| # FLOPs | 4.0 G | 7.7 G | 1.9 G | 8.8 G | 2.0 G | 579 M | 314 M | 220 M |

(a) Image classification on ImageNet

| Hyperparameter | Value |
|---|---|
| Epochs | 50 |
| Batch size | 16 |
| Data sampler | SSc-FBS |
| Warm-up iterations | 0 |
| Warm-up init. LR | NA |
| Max. LR | 0.02 |
| Min. LR | $10^{-4}$ |
| LR Annealing | cosine |
| Weight decay | $10^{-4}$ |
| Optimizer | SGD w/ momentum (0.9) |

| Hyperparameter | Value |
|---|---|
| Epochs | 32 |
| Batch size | 32,768 |
| Data sampler | MSc-VBS |
| Warm-up iterations | 2000 |
| Warm-up init. LR | $10^{-6}$ |
| Max. LR | $5^{-4}$ |
| Min. LR | $10^{-6}$ |
| LR Annealing | cosine |
| Weight decay | 0.2 |
| Optimizer | AdamW |

(b) Semantic segmentation on ADE20k    (c) CLIP (150M parameters) training on LAION-400M

Table 8: Hyper-parameters used for training models on different tasks. Here, SSc-FBS and MSc-VBS refers to single-scale fixed batch size and multi-scale variable batch size data samplers respectively (Mehta et al., 2022).

### F.2 BOUNDS FOR AUGMENTATION OPERATIONS

Table 9 shows the clipping bounds that `RangeAugment` uses to prevent training instability.

| Operation | Clipping bounds in `RangeAugment` | | Magnitude range in AutoAugment |
|---|---|---|---|
| | **Min.** | **Max.** | **(reference)** |
| Brightness | 0.1 | 10.0 | [0.1 - 1.9] |
| Contrast | 0.1 | 10.0 | [0.1 - 1.9] |
| Noise (std. dev) | 0 | 1.0 | – |

Table 9: Clipping bounds used in `RangeAugment` for preventing training instability. The range of magnitudes for augmentation operations in AutoAugment is also given as a reference.

