# OpenReview forum: "RangeAugment:  Efficient Online Augmentation with Range Learning"
_ICLR.cc/2023/Conference — Submitted to ICLR 2023_

### Official Review · Reviewer_39wu · 2022-10-24

**Confidence:** 5
**Correctness:** 3
**Technical Novelty And Significance:** 2
**Empirical Novelty And Significance:** 2
**Recommendation:** 3

**Clarity, Quality, Novelty And Reproducibility:**

Clarity: good.

Quality: the method is too simple, and many important aspects remain uncovered.

Novelty: very limited on a grid search on a single hyper-parameter.

Reproducibility: seems good, the method is simple anyway.

**Strength And Weaknesses:**

Strengths
1. The paper is easy to follow.
2. The studied problem (effective data augmentation) is of broad interest to the community.

Weaknesses
1. The proposed method is too simple. Of course, being simple is not a weakness, but the method does not show sufficient accuracy gain in experiments. Technically, Eq (2) tries to formulate in a generalized way, but the final solution is not "end-to-end" as claimed in the paper, but to grid-search the only parameter, \Delta. Besides, only studying the intensity is insufficient. There are many more aspects that need to be considered, such as the order of augmentation, the presence or absence of each augmentation operation, etc. In addition, the single function d (measured by PSNR) is only side evidence and I really doubt whether it can help to generate samples that help visual recognition (especially when I see the weak experimental results).

2. I went through the paper several times, but I cannot find the exact definition of L_ra. This does not prevent me from understanding the main idea, but the details are not clear.

3. The empirical study is trivial. I expect many technical tricks that can impact the effect of using DA, such as the use of knowledge distillation, e.g. [Wei et al., Circumventing outliers of autoaugment with knowledge distillation, ECCV20] suggested that the augmented data shall be checked by a teacher model to avoid less meaningful examples as shown in Fig 4. This technique was later used by DeiT [Touvron et al., Training data-efficient image transformers & distillation through attention, ICML21]. Of course, there are other tricks that can improve DA. Without considering these tricks, the experimental results are below satisfaction.

4. Regarding the three observations. Ob 1 (as used in AutoAugment and RandAugment) is mostly known. Ob 2 seems superficial -- the reason behind the observation is that small models often cooperate well with moderate DA (or similarly, moderate-scale datasets). In addition, the limited experiments in this paper are insufficient to validate the statement. Ob 3 is confusing to me -- RangeAugment only learns a single parameter Delta and a few parameter-free strategies (e.g. composition of 3 DA operations). Since the learned model is simple, it is naturally generalized, yet the improvement is also marginal.

5. Experimental results are weak. Please refer to the AA and RA papers for more thorough experiments. Most often, toy experiments are performed on CIFAR, and then transferred to ImageNet. Currently, the improvements on ImageNet-1K and ADE20K are mostly marginal, which cannot justify the effectiveness of RangeAugment.

**Summary Of The Paper:**

This paper presents an automatic learning approach for constructing training samples of data augmentation. For this purpose, a framework with three candidate operations is built and a grid search algorithm is used for finding the optimal Delta (for the intensity of DA). Experiments show limited accuracy gain on image classification and semantic segmentation.

**Summary Of The Review:**

The method is too simple and many aspects are not covered. In addition, the experiments are weak. I cannot suggest acceptance.

---

### Official Review · Reviewer_aoS6 · 2022-10-25

**Confidence:** 3
**Correctness:** 2
**Technical Novelty And Significance:** 2
**Empirical Novelty And Significance:** 2
**Recommendation:** 3

**Clarity, Quality, Novelty And Reproducibility:**

**Quality.** The original idea is interesting. I was not strongly convinced by the experiments and some of the details (such as the loss function used).

**Clarity.** I think there is definitely room for improvement in the writing and the plots.

**Originality.** Having differentiable augmentations and optimizing their parameters seemed novel to the best of my knowledge.

**Strength And Weaknesses:**

**Strengths**

1. I think there is merit to the idea of having differentiable augmentations and optimizing them.

**Weaknesses**

1. The experiments don't convey a strong sense of RangeAugment outperforming previous methods. E.g. in figure 5, RangeAugment seems to be underperforming compared to AutoAugment/RandAugment. Also, RangeAugment seems to be somewhat sensitive to the PSNR value choice.
2. I thought the authors would want to optimize the augmentation parameters based on the augmentation loss, but the paper suggests adding the augmentation loss term to the task loss. This did not make sense to me: Why would one want to optimize the model parameters with respect to the augmentation similarity loss?
3. Having an augmentation loss that computes how close the image is to the original image did not make sense to me. Why would this be a good loss function for determining the best augmentations?
4. I found the writing a bit difficult to follow - at times it felt like there was too much verbose information (e.g. section 4.2). I also found figures 5 & 7 hard to read - seemed crowded to me.

**Suggestions**

I could see one potentially exciting use-case for differentiable augmentations in sim2real scenarios: A common scenario is having labelled simulated training data (which often have perfect cameras), and some unlabelled real world images. People often augment the simulated training images to mimic real-world imperfections of the camera (which sometimes includes modelling the noise profile of the camera). Using differentiable augmentations, one can potentially look into optimizing augmentation parameters s.t. the features from the simulated training images are closer in distribution to real world images - basically omitting the need for manually modelling the camera noise profile.

**Summary Of The Paper:**

**Motivation**. Current automatic augmentation methods search over pre-determined range values for the strength of each augmentation. Ideally, the search should happen over continuous values for the range.

**Approach**. This paper proposes to differentiate with respect to the range of the augmentations. This is done by:
1. Using differentiable augmentation functions (brightness, contrast, additive gaussian noise)
2. Using the random variable re-parameterization trick to have differentiability with respect to the uniform distribution
3. For the loss, first the similarity between augmented and original image is computed (i.e., PSNR). Then, this value is compared to a desired value $\Delta$.


**Summary Of The Review:**

While this paper has an interesting idea in its core, I think the presentation and experiments need more work so that they clearly demonstrate the advantages of RangeAugment. The benefits of RangeAugment are not so clear to me in the current form of the paper.

---

> ### Comment · Reviewer_aoS6 · 2022-11-28
> **Thank you for the responses**
>
> Thank you for the responses
>
> > RangeAugment aims to learn model-specific policy.
>
> - In joint optimization, you're determining the augmentation policy (in part) based on the training loss. I don't conceptually understand using the training loss for determining augmentation values, whereas typically they are determined by monitoring validation performance (since the point of augmentation is to improve generalization).
>
> - In independent optimization, there is nothing model-specific about the augmentation getting learned. However, the augmentation will be dataset-specific and we can say we have a more interpretable "knob" over the augmentation ranges based on the PNSR value of the final augmented image (which I think is fair).
>
> >  it delivered similar validation performance to RandAugment/AutoAugment while having lower training accuracy. In other words, ResNet-50 trained with RangeAugment generalizes better than previous methods. We included the exact numbers in the below table.
>
> I disagree with the conclusion that RangeAugment generalizes better because its training accuracy is lower. The model that has a higher validation accuracy is expected to generalize better, which in this case is AutoAugment.

---

### Official Review · Reviewer_D2H4 · 2022-10-25

**Confidence:** 4
**Correctness:** 2
**Technical Novelty And Significance:** 2
**Empirical Novelty And Significance:** 2
**Recommendation:** 3

**Clarity, Quality, Novelty And Reproducibility:**

The writing is clear, but the experimental results are not satisfactory and not convincing.

**Strength And Weaknesses:**

**Strength**

The paper writing is clear, and the content is easy to follow. The observations and their related figure illustrations and analysis seem nice.

**Weaknesses**

The paper lacks lots of experimental evaluations to support its claim. For the shown image classification and semantic segmentation tasks, only ImageNet with several backbones and ADE20K with limited frameworks are tested. This is far from satisfactory, and readers can not be fully convinced by these results. Besides, the limited number of experimental results are also far from the current best solutions and SOTA performance. Further, experimental evaluations with more tasks like object detection are also recommended.

**Summary Of The Paper:**

This paper focuses on the problem of data augmentation. Previous search-based data augmentation methods like AutoAugment adopt a large set of operations that is not flexible and leads to sub-optimal solutions. The author proposes a method named RangeAugment to learn the range of magnitudes for each individual augmentation. The experimental evaluations are conducted on ImageNet for image classification and ADE20K for semantic segmentation, and the results seem ok.

**Summary Of The Review:**

The claims in the paper are not fully supported by the experimental evaluations. The paper seems not well prepared, and further investigation is needed to make it stronger.

---

### Official Review · Reviewer_Q5Gq · 2022-10-27

**Confidence:** 4
**Correctness:** 4
**Technical Novelty And Significance:** 4
**Empirical Novelty And Significance:** 3
**Recommendation:** 8

**Clarity, Quality, Novelty And Reproducibility:**

Clarity: The exposition is clear, but the clarity of the figures should be improved.

Quality: The quality is excellent.

Novelty: The proposed method for learning the range of magnitude of each augmentation operation is simple and novel.

Reproducibility: Sufficient implementation details are provided, but it would be great if the authors were willing to provide a reference implementation.

**Details Of Ethics Concerns:**

I don't see any ethics concerns.

**Strength And Weaknesses:**

Strength:
+ The paper's idea of replacing a fixed, manually defined range of data augmentation operations is novel and interesting.
+ It's surprising that by only using three operations (brightness, contrast, noise), the proposed augmentation achieves state-of-the-art performance like other methods that use many more operations (e.g., AutoAugment, RandAugment).
+ The evaluation is thorough. The experiments answer many questions regarding generalization across different model architectures and tasks.

Weakness:
- The font sizes of many of the figures are too small. It's hard to read (without zooming in on a computer).
- I am curious whether the range of magnitude idea can be applied to other types of data augmentation beyond changing brightness/contrast. For example, it would be interesting to see how RangeAugment can be applied to geometric augmentation (zoom, cropping, rotation).

**Summary Of The Paper:**

The paper proposes a new data augmentation technique for visual recognition tasks. The core idea is to replace the fixed, manually defined ranges for data augmentation operations (such as brightness and contrast) with a learned, adaptive policy for specifying the magnitude of the operation range. The learned policy is model-specific and task-specific. The paper shows an extensive evaluation of ImageNet across different networks. The results show that the proposed RangeAugment achieves similar performance as the state-of-the-art AutoAugment and RandAugment but requires significantly fewer data augmentation operations. The paper also demonstrates results on other tasks such as semantic segmentation and contrastive learning.

**Summary Of The Review:**

The paper presents a novel data augmentation technique. Learning the range of magnitude for each data augmentation supports learning model-specific and task-specific policies. The method is simple and effective (as shown in the experiments). While only simple augmentation operations are shown (brightness, contrast, noise) in this paper, I think the proposed data augmentation method could benefit the community.

---

### Decision · Program_Chairs · 2023-01-20

**Decision:**

Reject

**Justification For Why Not Higher Score:**

experimental results are marginal/inferior to the baselines

**Justification For Why Not Lower Score:**

n/a

**Metareview: Summary, Strengths And Weaknesses:**

This paper proposes a method to learn the range of continuous augmentation operation using auxiliary loss based on image similarity.

Three reviewers suggested reject, while one reviewer recommended accept. Although no consensus was reached even during the discussion period, positive reviewers did not strongly support accept.

This AC also carefully reviewed the paper at the request of the authors, but agreed with the opinions of the reviewers who argued for rejection. In particular, unlike the claim in the abstract that it can be applied to composite augmentation, it was actually applied only to brightness or contrast, and they only shared future plans for other augmentations. In addition, despite the existence of attractive experimental results in the appendix, the biggest weakness of this paper is that the experimental results of the main experimental section are marginal, or that the validation performance is rather inferior to the baselines such asRandAug/AutoAug. The fact that a small number of augmentations can produce similar effects to the baseline is less attractive when considering the original purpose of augmentation. I think it is necessary to show in the main experiment section that the proposed method can achieve superior performance with similar number of augmentations.